# Effects of Liupao Tea with Different Years of Aging on Glycolipid Metabolism, Body Composition, and Gut Microbiota in Adults with Obesity or Overweight: A Randomized, Double-Blind Study

**DOI:** 10.3390/foods14050866

**Published:** 2025-03-03

**Authors:** Yuyang Wang, Qiang Hu, Botian Chen, Defu Ma

**Affiliations:** Department of Social Medicine and Health Education, School of Public Health, Peking University Health Science Center, Beijing 100191, Chinaqianghu@stu.pku.edu.cn (Q.H.);

**Keywords:** Liupao tea, weight management, lipid profile, gut health, randomized trial

## Abstract

Background: Liupao tea (LPT) is a traditionally fermented dark tea from Guangxi, China and the effects of different aging periods of LPT on metabolic health remain inadequately explored. Methods: This randomized, double-blind, longitudinal study enrolled 106 adults with obesity or overweight who were assigned to consume LPT of different ages over a 90-day period. Participants were randomly divided into four groups, each consuming LPT that had been aged for 1 year, 4 years, 7 years, or 10 years. The metabolic parameters, body composition, and gut microbiota were assessed at baseline and after the 90-day intervention. Results: All LPT groups experienced significant reductions in systolic blood pressure (SBP) and diastolic blood pressure (DBP), with the 10-year-aged group showing the most notable SBP decrease (*p* < 0.001). Total cholesterol (TC) and low-density lipoprotein cholesterol (LDL-C) levels decreased significantly in the 1-, 4-, and 10-year-aged groups (*p* < 0.05), while high-density lipoprotein cholesterol (HDL-C) increased in the 7-year-aged group (*p* < 0.05). Body weight, body fat mass (BFM), body mass index (BMI), waist circumference (WC), body fat percentage (BFP), and visceral fat area (VFA) significantly declined across all groups (*p* < 0.05). Gut microbiota analysis showed changes in specific genera, though overall diversity remained stable. No significant differences were found in metabolic or microbiota outcomes between the different aged groups. Conclusions: LPT consumption effectively improves blood pressure, lipid profiles, and body composition in adults with obesity without adverse liver effects. The aging duration of LPT does not significantly alter these health benefits, challenging the belief that longer-aged LPT is superior.

## 1. Introduction

Metabolic diseases, including obesity, diabetes, cardiovascular diseases, and fatty liver, have emerged as major global public health concerns [1,2]. According to the latest data from the Non-Communicable Disease Risk Factor Collaboration (NCD-RisC) and the World Health Organization (WHO), the global prevalence of obesity has shown a steady upward trend, affecting approximately 880 million adults (504 million women and 374 million men) affected by obesity in 2022 [3]. Obesity and its associated metabolic disorders significantly increase the incidence and mortality of cardiovascular diseases, diabetes, stroke, and other conditions, while also imposing a substantial socioeconomic burden [4]. Recent reports indicate that in 2022, the global economic loss caused by overweight and obesity, including healthcare costs and productivity losses, accounted for 0.05% to 2.42% of global GDP across various countries [5]. Therefore, identifying effective interventions to prevent and manage obesity and its related metabolic diseases has become a critical issue in global health.

Lifestyle improvements, especially dietary interventions, have been shown to play a crucial role in the prevention and management of metabolic diseases [6,7,8]. Fermented teas, through microbial fermentation, yield unique bioactive compounds that potentially modulate metabolic health. As a specific type of black tea, Pu-erh tea has demonstrated the ability to alter body composition and energy efficiency in mice fed a high-fat diet. It has been shown to alleviate metabolic endotoxemia, reduce systemic and multi-organ inflammation, and improve glucose and lipid metabolic disorders [9]. The theabrownins present in Pu-erh tea have been found to significantly reduce serum total cholesterol (TC) and low-density lipoprotein (LDL) levels by inhibiting hepatic cholesterol synthesis and promoting lipid catabolism [10,11]. Ya’an Tibetan tea, a traditionally fermented tea, was demonstrated by Ye et al. to significantly improve weight management and fat accumulation in HFD-induced obese rats. It enhances antioxidant enzyme activity, reduces inflammation, and reverses HFD-induced hepatic damage [12]. Kombucha, a non-alcoholic fermented tea beverage produced through symbiotic bacterial and yeast fermentation, has been reported to exert beneficial effects on metabolic health. According to a literature review by Costa, kombucha consumption can significantly mitigate oxidative stress and inflammation, enhance hepatic detoxification processes, and provide potential benefits for the management and treatment of obesity and its related comorbidities [13].

Liupao tea (LPT), a type of dark tea produced in Guangxi, China, is known for its unique fermentation process and aging procedure, which endow it with distinctive chemical components and bioactivities [14,15]. Compared to green tea, LPT has higher levels of alkaloids and fatty acids, and contains relatively higher amounts of ellagic acid, hypoxanthine, and theobromine [16]. Ding et al. found that LPT extract significantly alleviated hyperglycemia, insulin resistance, and dyslipidemia in streptozotocin-induced diabetic rats, with its blood glucose-lowering effect comparable to that of metformin when administered at high doses [17].

Studies have shown that bioactive compounds in tea, such as polyphenols, theaflavins, and tea polysaccharides, possess multiple health effects, including antioxidant, anti-inflammatory, blood glucose-lowering, lipid-lowering, blood pressure-regulating, and gut microbiota-modulating activities [18,19,20,21]. In traditional Chinese medicine, LPT is believed to have “lipid-lowering, spleen-strengthening, and digestive-promoting” effects [22]. The microbial metabolites and chemical components (such as tea polyphenols, free amino acids, and water-soluble sugars) produced during the fermentation process of LPT change over time, and these changes play an important role in metabolic regulation [23,24]. However, there is still limited research on the effects of different aging years of LPT on body composition, glycolipid metabolism, and gut microbiota, particularly the potential effects of different aging years of LPT in populations with metabolic abnormalities such as obesity, hyperglycemia, and dyslipidemia.

This study investigates the effects of LPT, aged for different amounts of time (1, 4, 7, and 10 years), on glycolipid metabolism, body composition, and gut microbiota in adults with obesity or overweight through a longitudinal analysis. It is hypothesized that LPT consumption, regardless of length of aging, will demonstrate significant improvements in metabolic outcomes, including blood pressure, lipid profiles, and glucose regulation, while simultaneously reducing body fat accumulation and reshaping gut microbiota composition. Consequently, this research seeks to elucidate the mechanisms by which LPT modulates metabolic health through gut microbiota changes, thereby providing novel dietary intervention strategies for the management of metabolic disorders.

## 2. Materials and Methods

### 2.1. Study Design

This is a randomized, double-blind, longitudinal study with a 90-day intervention period, exploring the effects of LPT aged for different numbers of years on glycolipid metabolism, body composition, and gut microbiota in adults with obesity or overweight. This study was approved by the Biomedical Ethics Committee of Peking University (NO.IRB00001052-23203) and registered with the Chinese Clinical Trial Registry (Trial Registration number: ChiCTR2400089348). Figure 1 illustrates the study flow, from recruitment to data analysis.

### 2.2. Participant Recruitment and Eligibility Criteria

The recruitment process began on 1 April 2024, with the posting of project promotional posters in the Xueyuan Road community in Haidian District, Beijing, and concluded on 30 April 2024. The inclusion criteria were as follows: (1) age between 18 and 70 years; (2) BMI ≥ 24.0 kg/m^2^; (3) no severe liver, kidney, gastrointestinal diseases, or diabetic complications; (4) no use of medications or dietary supplements to lower blood lipids, blood glucose, or body weight in the past three months, and agreement not to use such substances during the study; (5) voluntary participation after receiving a detailed explanation of the project and signing the informed consent form.

Exclusion criteria included: (1) pregnant, breastfeeding, or pre-conception women; (2) serious heart, liver, kidney, hematologic diseases, or psychiatric disorders; (3) use of lipid-lowering, glucose-lowering, or weight-loss medications or supplements in the last three months that may affect the study outcomes; (4) excessive smoking or alcohol consumption; (5) special dietary habits such as vegetarianism or ketogenic diets (high-fat, low-carb); (6) inability to follow study requirements; (7) allergies to tea components. Informed consent was obtained from all participants under the guidance of the researchers to ensure that each participant fully understood the main content of the study, its associated risks, and potential benefits.

### 2.3. Sample Size Estimation

The sample size for this study was estimated based on previous research on dark tea, where BMI decreased from 28.67 ± 2.91 kg/m^2^ to 26.86 ± 3.17 kg/m^2^. Using a longitudinal design, with a significance level (α) of 0.05 and a power (1 − β) of 80%, the required sample size per group was calculated to be 24 participants. Accounting for a 10% dropout rate, the final sample size for each group was 26, resulting in a total of 104 participants [25].

### 2.4. Intervention Materials

The LPT samples used in the experiment all originated from Zhaoping County, Guangxi Zhuang Autonomous Region. During the aging process, the samples were stored in a constant temperature and humidity warehouse with a temperature range of 20–25 °C, relative humidity of 60–65%, ensuring proper air circulation and no direct sunlight exposure. The LPT used in the experiment was provided free of charge by Guangxi Zhaoping County General Mountain Agricultural Technology Co., Ltd. (Hezhou, China). The main water-soluble components identified in these LPT samples for the different years of aging are summarized in Appendix A.

### 2.5. Randomization and Blinding

Participants were randomly assigned into four groups based on the aged years of LPT: 1-year, 4-year, 7-year, and 10-year-aged tea groups. Randomization was performed by a statistician not involved in the trial using computer-generated random numbers to ensure impartiality and randomness in group assignment. The randomization list was then handed over to the sample packager, who packaged the tea samples (6 g per bag) according to the assigned groups while ensuring uniformity in the external packaging of each tea bag. The study followed a double-blind design, where both participants and researchers were unaware of the specific tea aging groups to avoid bias. Participants were notified to collect the pre-packaged tea from the study office on the 2nd day before the intervention, as well as on the 28th and 58th days of the intervention, and their names and collection times were recorded.

### 2.6. Intervention and Adherence Monitoring

During the intervention, participants consumed 6 g of LPT (one tea bag) daily. The tea was brewed using a standardized 300 mL teapot, with 250 mL of boiling water for a 30–45 s infusion, followed by immediate consumption once the tea had cooled to a warm temperature. Participants were allowed to re-brew the tea until the color of the tea became pale. To optimize the intervention’s effectiveness, participants were instructed to drink the tea 1–2 h after each meal, and to complete their daily consumption before 3 p.m. to minimize potential disruption to nighttime sleep.

To improve adherence, a dedicated WeChat group was created where participants uploaded daily photos of their tea consumption as proof of compliance and kept a personal tea diary. Research staff monitored the group daily, providing reminders to participants who missed their tea consumption. If participants experienced any intolerance or adverse effects, they were instructed to immediately stop drinking the tea and contact the researchers. Participants who failed to comply with the intervention protocol or who could not be reached for three consecutive days were withdrawn from the study, and the reason for discontinuation or dropout was documented. Throughout the study, participants were advised to maintain their usual diet, exercise, and sleep routines without making any significant lifestyle changes. Adherence to the intervention was closely monitored, with daily follow-up and reminders provided to participants. Any non-compliance or adverse events were recorded and addressed promptly. If participants were lost to follow-up or dropped out of the study, the reasons for discontinuation were documented.

### 2.7. Questionnaire Survey and Clinical Parameter Measurement

At baseline, demographic data, including gender, age, marital status, education level, smoking habits, alcohol consumption, and family history of chronic diseases, were collected via questionnaires. The following measurements were taken at baseline and after 90 days of intervention. The clinical parameters collected in this study include blood pressure, lipid metabolism indicators, glucose metabolism indicators, and liver function markers. Before blood pressure measurement, participants were required to sit quietly in a calm environment for 5 min. The upper arm was exposed and kept level during measurement. Blood pressure was measured three times, and the average of the three measurements was taken as the final value. On the morning of the test, approximately at 8:00 A.M., fasting venous blood (after fasting for more than 8 h) was drawn from participants into centrifuge tubes. After centrifugation and aliquoting, the samples were stored in a −80 °C ultralow freezer until testing. Fasting blood glucose (FBG) was measured using the hexokinase method, hemoglobin A1c (HbA1c) was measured by ion-exchange chromatography, and fasting insulin levels (INS) were measured using chemiluminescent immunoassay. TC, triglycerides (TG), high-density lipoprotein cholesterol (HDL-C), low-density lipoprotein cholesterol (LDL-C), apolipoprotein A1 (APOA1), and apolipoprotein B (APOB) were assessed by enzyme methods. To evaluate the safety of LPT intervention, key biomarkers of liver health, including total bilirubin (T-BIL), alanine aminotransferase (ALT), aspartate aminotransferase (AST), serum alkaline phosphatase (ALP), serum creatine kinase (CK), and serum lactate dehydrogenase (LDH), were also measured. All participant samples were collected by professional medical staff at Beijing Meinian Health Examination Center and sent to a specialized laboratory for analysis using an automated biochemical analyzer.

### 2.8. Body Composition Measurement

Height was measured with participants barefoot and standing upright, with their back against a wall. A right-angle board was placed on the participant’s head, and the measurement was repeated twice, with the average of the two used as the final height. Waist circumference (WC) was measured with a standard soft tape at 1 cm above the navel. The measurement was repeated twice, and if the difference between the two was less than 0.5 cm, the average value was taken as the final waist circumference. Body composition was assessed using the IOI353 body composition analyzer (Jawon Medical Co., Ltd., Seoul, Republic of Korea). After verifying participant details (enrollment number, gender, and age) on the device, participants were instructed to remove their outerwear, stand barefoot on the analyzer’s electrodes, and grip the hand electrodes at a 30-degree angle to their body. Participants were instructed to remain still during the measurement. Body mass index (BMI) is calculated by dividing the subject’s weight (kg) by the square of their height (m). The analyzer measured body weight, body fat mass (BFM), muscle mass (MM), body fat percentage (BFP), lean body mass (LBM), and visceral fat area (VFA).

### 2.9. DNA Extraction and Amplification Library Construction

Total genomic DNA was extracted from the samples using the SDS method to ensure that the quality of the extracted DNA was suitable for subsequent amplification and sequencing. Specific primer pairs (515F and 806R) were used to amplify the V3-V4 regions of the 16S rRNA gene. The PCR reaction mixture contained 15 µL of Phusion^®^ High-Fidelity PCR Master Mix (New England Biolabs, Inc., Ipswich, MA, USA), 0.2 µM of each primer, and 10 ng of genomic DNA. The PCR conditions were as follows: an initial denaturation at 98 °C for 1 min, followed by 30 cycles at 98 °C for 10 s, 50 °C for 30 s, and 72 °C for 30 s, with a final extension at 72 °C for 5 min. After electrophoresis on a 2% agarose gel to confirm the PCR products, the samples were purified using magnetic beads and mixed in equal proportions. The constructed libraries were evaluated for quality using a Qubit^®^ 2.0 Fluorometer (Thermo Fisher Scientific, Inc., Waltham, MA, USA) and the Agilent Bioanalyzer 2100 system (Agilent Technologies, Inc., Santa Clara, CA, USA). The qualified libraries were sequenced on an Illumina NovaSeq platform (Illumina, Inc., San Diego, CA, USA), generating 250 bp paired-end reads.

### 2.10. Bioinformatics Analysis and Diversity Analysis

After splitting the sequencing data based on barcode and primer sequences, paired-end reads were assembled using FLASH (Version 1.2.11, https://ccb.jhu.edu/software/FLASH/, accessed on 15 October 2024), generating the raw tag sequences. The raw data were then quality-filtered using fastp (Version 0.23.1) to obtain clean tags. The effective tags were obtained by removing chimeric sequences through alignment with the Silva database (Version 2.15.0) using Vsearch. Denoising of the effective tags was performed using the DADA2 algorithm or the deblur module in QIIME2, resulting in amplicon sequence variants (ASVs). Taxonomic annotation of the ASVs was performed using the Silva138.1 database for 16S rRNA gene sequences, with supplementary taxonomic information retrieved from the NCBI database when necessary. Based on the normalized data, both α-diversity and β-diversity analyses were performed to evaluate the diversity within and between the microbial communities across the samples. Based on the species annotation and abundance information at the genus level for all samples, the top 35 genera were selected. Clustering was performed at the species level using their abundance data in each sample to identify the relative abundance of species across the groups.

### 2.11. Statistical Analysis

For the basic demographic information, clinical parameters, and body composition indicators of the participants, continuous variables that followed a normal distribution are expressed as mean ± standard deviation (SD), with group differences assessed using t-tests or one-way analysis of variance (ANOVA). Categorical variables are presented as frequencies (n) and percentages (%), with group differences compared using the chi-square test. If the ANOVA results show significant differences (*p* < 0.05), Bonferroni correction or the least significant difference (LSD) method will be used to conduct post hoc pairwise comparisons. The missing data of dropouts were handled using multiple imputation by chained equations (MICE) and the robustness of the results was assessed using intention-to-treat (ITT) analysis.

Alpha diversity was calculated using indices such as Shannon, Simpson, and Chao1, and was evaluated using QIIME2 software to assess the richness and diversity of microbial communities within the samples. Beta diversity was used to compare microbial community compositions between different samples. Based on species annotation results and feature sequence abundance, sequences within the same taxonomic group were merged to generate a species abundance table (profiling table). Furthermore, UniFrac distances were calculated based on the phylogenetic relationships between feature sequences, including both unweighted and weighted UniFrac distances. These distances, which are calculated based on the evolutionary information of microbial sequences between samples, formed a distance matrix. This UniFrac distance matrix, whether weighted or unweighted, was then transformed into a new set of orthogonal axes, with the first principal coordinate representing the factor of maximum variation, the second principal coordinate representing the second largest variation factor, and so on. Beta diversity differences were visualized using principal coordinates analysis (PCoA), revealing differences between samples (or groups). Two-dimensional PCoA results were displayed using the ade4 and ggplot2 packages (version 2.15.3) in R software (version 4.1.3). To evaluate the significance of differences in community structure between groups, the adonis and anosim functions in QIIME2 were used for analysis. The Mann–Whitney U test was used to compare significantly different species at each taxonomic level (phylum, class, order, family, genus, species) before and after the intervention across the LPT of different aging years. To investigate the functional roles of microbial communities and identify differences between groups, PICRUSt2 software (Version 2.1.2-b) was used for functional annotation. KO (KEGG Orthology) functions were ranked by relative abundance at baseline and follow-up, and the top 10 KO IDs were selected. The Wilcoxon signed-rank test was used to compare changes within the same samples over time.

All statistical analyses were performed using R statistical software version 4.1.3, with statistical significance defined as *p* < 0.05.

## 3. Results

### 3.1. Basic Characteristics of Participants

Table 1 showed the comparison of baseline characteristics between the total population and different aged intervention groups. The total sample size was 106 participants, including 26 in the 1-year-aged group, 28 in the 4-year-aged group, 26 in the 7-year-aged group, and 26 in the 10-year-aged group. The gender distribution in the total population was 26.4% male and 73.6% female. There were no significant differences between the groups in terms of age, gender, marital status, education level, monthly income, alcohol consumption, and tea-drinking habits. Among the participants, 79.2% had T2DM, 67% had hypertension, and 61.3% had dyslipidemia. No significant differences were observed across the groups in the distribution of these medical history (*p* = 0.100, *p* = 0.756, and *p* = 0.526).

### 3.2. Effects of LPT with Different Aged Years on Metabolic Parameters

Table 2 presented the intervention effects of LPT with different aged years on metabolic parameters. After 90 days of tea intervention, SBP significantly reduced in all four groups. In the 10-year-aged group, SBP dropped from 131.32 ± 16.08 mmHg to 121.16 ± 15.16mmHg (*p* < 0.001). DBP showed a significant decline in the 1-year-aged, 4-year-aged, and 10-year-aged groups (*p* < 0.05). TC in the 1-year-aged group showed a marked reduction (*p* < 0.001), from 5.10 ± 1.08mmol/L to 4.87 ± 0.88 mmol/L. LDL-C decreased significantly in the 1-year-aged, 4-year-aged, and 10-year-aged groups (*p* < 0.05), while HDL-C notably increased in the 7-year-aged group (*p* < 0.05). APOA1 was significantly lower in all four groups (*p* < 0.05). FBG levels in the 4-year-aged group were significantly reduced, from 5.75 ± 1.19 mmol/L to 5.50 ± 1.07mmol/L (*p* < 0.05). Additionally, LPT had no significant effect on most liver health biomarkers (T-BIL, ALT, AST, ALP, CK, LDH), indicating minimal impact on liver function and confirming its good safety profile (Appendix A).

### 3.3. Effects of LPT with Different Years of Aging on Body Weight and Composition

After 90 days of intervention, the 1-year-aged group, 4-year-aged group, 7-year-aged group and 10-year-aged group exhibited significant reductions in weight and BFM (*p* < 0.05). In the 4-year-aged group, weight was reduced from 69.00 ± 11.96 kg to 66.70 ± 10.77 kg (*p* < 0.001), and BFM dropped from 21.88 ± 5.64 kg to 19.54 ± 5.59 kg (*p* < 0.001). The 1-year-aged group showed significant increases in LBM and MM (*p* < 0.05). BMI was significantly lower in the 1-year, 4-year, 7-year and 10-year-aged groups (*p* < 0.05), and WC was significantly reduced post-intervention in the first three groups (*p* < 0.001). Additionally, BFP and VFA were significantly reduced in the 1-year, 4-year, 7-year, and 10-year-aged groups (*p* < 0.001) (Table 2). Overall, LPT demonstrated significant effects in reducing body fat, visceral fat, and abdominal fat accumulation.

### 3.4. Comparison of the Effects of LPT with Different Years of Aging on Metabolic and Body Composition Parameters

The comparison of the effects of 1-year, 4-year, 7-year, and 10-year-aged LPT groups on metabolic parameters and body composition was conducted by calculating the differences between follow-up and baseline levels. The results revealed no significant differences among the groups in reducing blood pressure (SBP, DBP), lipid metabolism (TC, TG, HDL-C, LDL-C), glucose metabolism (HbA1c, FBG), or changes in body weight and composition (weight, BFM, BMI, BFP), with *p*-values for intergroup differences exceeding 0.05 (Appendix A).

### 3.5. Characteristics of Gut Microbiota in LPT Groups with Different Years of Aging 

The principal coordinate analysis (PCoA) of beta diversity revealed no significant differences in gut microbiota structure between baseline and follow-up across all four LPT groups with different years of aging (Figure 2a). Specifically, no significant differences were observed in either PCoA1 or PCoA2 for any of the groups, with *p*-values ranging from 0.343 to 0.865. Similarly, the analysis of alpha diversity using the Shannon, Simpson, and Chao1 indices showed no significant changes in microbial diversity before and after the intervention in all four groups (Figure 2b). These results indicate that the gut microbiota structure and diversity remained stable over time across all experimental groups.

At the phylum level, the gut microbiota composition remained highly consistent across the 1-year, 4-year, 7-year, and 10-year-aged groups, with Firmicutes, Bacteroidota, Proteobacteria, Actinobacteriota, and Verrucomicrobiota being the predominant phyla (Figure 3a). However, a notable reduction in the relative abundance of Firmicutes was observed post-intervention in all four groups. In contrast, Bacteroidota exhibited a decrease in relative abundance in the 1-year, 4-year, and 7-year-aged groups, while the 10-year-aged group showed a significant increase in Bacteroidota abundance, accompanied by a rise in the Bacteroidetes/Firmicutes ratio. At the genus level, the gut microbiota composition remained consistent across all four groups, with Bacteroides being the dominant genus and exhibiting the highest relative abundance in the 1-year, 4-year, 7-year, and 10-year-aged groups. Prevotella_9 and Faecalibacterium followed as the next most abundant genera. Post-intervention, the 1-year-aged group showed a significant increase in the relative abundances of Escherichia−Shigella and Bifidobacterium compared to baseline levels. Beyond the top three dominant genera, the 4-year-aged group exhibited elevated abundances of Bifidobacterium and Megamonas, whereas the 10-year-aged group had higher relative abundances of Blautia and Roseburia (Figure 3b).

At the genus level, significant differences were observed in the gut microbiota composition across the four LPT groups with different years of aging before and after the intervention. In the 1-year-aged group, the relative abundance of Stenotrophomonas, Rothia, and Delftia decreased significantly, while Intestinibacter and Eubacterium halli_group increased notably (Figure 4a). In the 4-year-aged group, there were significant increases in Peptostreptococcus, Dubosiella Faecalibaculum, Allobaculum, Eubacterium_xylanophilum_group, Monoglobus, Eubacterium_ventriosum_group, and Escherichia-Shigell after the intervention (Figure 4b). As for the 7-year-aged group, Pseudomonas showed a significant decrease, while UCG.007 and Oxalobacter, displayed notable increases (Figure 4c). The 10-year-aged group exhibited a significant reduction in Erysipelotrichaceae_UCG-003, FukuN18_freshwater_group, and Polynuceobacter (Figure 4d). Overall, the LPT intervention exhibited varying effects across the different years of aging, significantly altering the relative abundance of certain genera in the gut microbiota. Functional prediction analysis using PICRUSt2 revealed no significant differences in the top ten functional features among the differently aged LPT groups (Appendix A).

## 4. Discussion

### 4.1. Main Findings and Comparison with Prior Studies

This study systematically assessed the effects of LPT aged for various years on metabolic parameters, body composition, and gut microbiota, revealing for the first time its beneficial roles in regulating blood pressure, lipids, glucose, and body weight in a human population.

Similar to our study, epidemiological studies associated black tea consumption with reductions in SBP of 1.79–5.31 mmHg and DBP of 0.47–1.02 mmHg [26]. Li et al. demonstrated in a rat model that Pu-erh tea extract enhanced the antihypertensive effects of low-dose nifedipine, achieving blood pressure reductions comparable to high-dose nifedipine alone [27]. LPT is rich in secondary metabolites, particularly polyphenols, which significantly increase the number of functionally active circulating angiogenic cells, thereby counteracting the impairment of flow-mediated dilation caused by fat intake [28,29].

LPT also positively influenced lipid and glucose metabolism. TC levels significantly decreased in the 1-year-aged group, and HDL-C levels significantly increased in the 7-year-aged group, consistent with animal studies. Huang et al., using a hyperlipidemia Sprague–Dawley rat model, found that TC and TG levels in various dosage LPT groups were significantly lower than in the control group, with LDL-C levels also reduced. LPT was more effective than green tea in lowering TC, TG, and LDL-C levels [16]. Wu et al. reported that LPT extract reduced body weight in obese mice, lowered liver TC, TG, and LDL-C levels, and increased HDL-C levels, indicating that LPT alleviates lipid metabolic disorders induced by a high-fat diet [30]. Tannic acid, catechins, and naringenin are identified as key active components of LPT in mitigating hyperlipidemia [24]. Yang et al., using a non-alcoholic fatty liver disease mouse model, found that LPT extract significantly reduced the number and size of hepatic lipid droplets, with therapeutic effects comparable to simvastatin [31]. Regarding blood glucose, FBG significantly decreased in the 4-year-aged group. This aligns with Wu et al.’s findings in animal models, where LPT intervention accelerated the reduction in blood glucose following oral glucose administration, confirming that aged LPT improves glucose tolerance in high-fat diet-induced obese mice [32]. Ding et al. discovered that high-dose LPT extract treatment for six weeks significantly reduced 2 h postprandial blood glucose (2h-PBG) and FBG levels in diabetic rats, exhibiting a dose-dependent hypoglycemic effect [17]. As a post-fermented tea, LPT undergoes long-term microbial fermentation, during which polyphenolic compounds such as tea polyphenols and catechins oxidize and polymerize to form higher molecular weight derivatives like theaflavins. These polyphenols activate the AMP-activated protein kinase (AMPK) pathway, promoting β-oxidation of fatty acids, increasing fat decomposition, and enhancing energy expenditure [33]. Additionally, AMPK activation inhibits fat synthesis-related enzymes (e.g., fatty acid synthase and acetyl-CoA carboxylase), reducing fat synthesis and accumulation, and lowering serum TC and LDL-C levels [34].

All LPT groups exhibited significant reductions in body weight and BFM post-intervention, with the 4-year-aged group showing the most pronounced decreases. BMI, BFP and VFA significantly declined across all LPT groups, and WC also significantly decreased in the 1-year-aged, 4-year-aged, and 7-year-aged groups. Wu et al. found that LPT extract reduced body weight and significantly alleviated liver damage in mice [30]. Li et al. reported that Sichuan black tea significantly reduced weight gain in high-fat diet mice and markedly lowered serum lipid profiles and atherosclerosis indices [35]. Additionally, studies have shown that black tea reduces weight gain and fat accumulation, improves glucose tolerance, alleviates metabolic endotoxemia, and regulates the mRNA expression levels of lipid metabolism-related genes, resulting in weight loss of 15–16% and body fat reduction of 38–44% [36]. Notably, the 1-year-aged group also showed significant increases in LBM and MM, suggesting that LPT may not only reduce fat but also promote muscle growth and maintenance. Previous research identified epigallocatechin as a major bioactive compound in black tea extract that alleviates palmitic acid-induced muscle atrophy by regulating mitochondrial function in C2C12 cells [37]. Catechins in tea maintain the dynamic balance between protein synthesis and degradation and promote mitochondrial energy metabolism, thereby mitigating muscle atrophy and maintaining muscle homeostasis [38].

Regarding gut microbiota, all groups showed a significant decrease in the relative abundance of Firmicutes post-intervention, while Bacteroidota decreased in the 1-year-aged, 4-year-aged, and 7-year-aged groups, with only the 10-year-aged group showing a significant increase in Bacteroidota and Bacteroidetes/Firmicutes ratio. The Bacteroidetes/Firmicutes ratio is a crucial indicator of gut microbiota health. Ding et al. found that LPT extract beneficially modulated diabetes-induced gut microbiota dysbiosis by increasing the Bacteroidetes/Firmicutes ratio [17]. Zhou et al., using a rat model of irritable bowel syndrome, observed that LPT increased the Bacteroidota/Firmicutes ratio and significantly reconstructed the microbial community to alleviate irritable bowel syndrome [39]. At the genus level, Bacteroides, the predominant genus, had the highest relative abundance in all aged LPT groups, followed by Prevotella_9 and Faecalibacterium. Similar to the 10-year-aged group, previous studies showed that LPT extract increases Bacteroides abundance in high-fat diet mouse models [40]. Polysaccharides in LPT more effectively promote butyrate production and increase Bacteroides abundance [41]. The Bacteroides genus plays a vital role in decomposing complex polysaccharides, synthesizing vitamins, and regulating the host immune system [42].

Our study also found that the 1-year-aged LPT intervention significantly increased the relative abundance of Intestinibacter and the Eubacterium_hallii group in the gut microbiota, consistent with Yang et al.’s findings. Eubacterium_hallii is an efficient butyrate and propionate producer, and its increased abundance is closely associated with improvements in metabolic parameters, such as reduced blood glucose, improved lipid profiles, and decreased body fat [43]. In the 4-year-aged group, several gut genera significantly increased post-intervention, and gut microbial diversity also improved. Notably, the Eubacterium_xylanophilum and Eubacterium_ventriosum groups exhibited the most significant changes; these bacteria produce short-chain fatty acids by digesting dietary fiber, exerting anti-obesity effects [44]. Additionally, Dubosiella abundance significantly increased in the 4-year-aged group. Studies have shown that Dubosiella, as a next-generation probiotic, can inhibit the mTOR pathway, thereby reducing blood lipids and enhancing insulin sensitivity [45]. We also observed a significant increase in Allobaculum abundance in the 4-year-aged group. Zheng et al. found that Allobaculum abundance positively correlates with the expression of ANGPTL4, a key regulator of lipid metabolism [46]. In the 10-year-aged group, the abundance of Erysipelotrichaceae UCG-003 significantly decreased. Previous studies indicate that increased Erysipelotrichaceae abundance is associated with a higher risk of developing irritable bowel syndrome and is related to intestinal inflammation processes [47]. The top ten functional features by relative abundance did not show significant differences, which may be related to the insufficient sample size and the instability of functional predictions. However, these limitations highlight the need for future studies to employ shotgun metagenomic sequencing, which can provide a more comprehensive understanding of the microbial community’s functional potential by directly sequencing the total DNA of the microbiota.

Despite traditional Chinese beliefs that longer-aged LPT possesses superior quality, our results indicate that LPT samples aged for different durations do not exhibit significant differences in improving blood pressure, blood lipids, blood glucose, body weight, or body composition parameters. Previous research has found that the antioxidant capacity of LPT follows a parabolic trend with storage time, peaking at five years of storage. Beyond this threshold, the antioxidant activity of LPT may decline, potentially due to reductions in catechins and glycosylation of flavonoids [24]. While Wu et al. found that the 10-year-aged group had better anti-obesity effects than the 1-year-aged group in obese mice, our study did not observe significant differences in the health effects of LPT across the different aging durations [32]. This may be related to the dynamic changes in LPT’s components. As the aging time extends, antioxidant components such as tea polyphenols and catechins gradually decrease, while flavonoids and soluble dietary fiber show dynamic changes or increasing trends. The synergistic effects of these components may, to some extent, balance the impact of the reduction in individual components on metabolic health. Additionally, individual differences and limitations in experimental design may also mask potential differences in the effects of aging time on metabolic health. While aging time can greatly affect the composition and flavor of LPT, its influence on human metabolic health is likely more complex and not determined solely by aging duration. Future studies should employ larger sample sizes and extended intervention periods to comprehensively explore how these chemical changes impact metabolic health.

### 4.2. The Strengths and Limitations of the Study

The strength of this study lies in its comprehensive assessment of the impact of LPT on metabolic health and gut microbiota based on a population experiment. By including LPT aged for different numbers of years, the study provides a unique perspective for analyzing the variations in their health effects. This study has several limitations. Firstly, due to practical and ethical considerations, the study design lacked a control group, as all participants received LPT interventions of different aging durations. This absence makes it impossible to rule out time effects or other external factors influencing the results, thereby limiting our ability to establish the independent health effects of LPT. Consequently, it remains uncertain whether the observed metabolic improvements are solely attributable to LPT consumption. Secondly, the study did not thoroughly investigate and record participants’ dietary and physical activity habits. Although participants were instructed to maintain their usual diet and exercise routines, the lack of systematic measurement may have allowed minor changes in dietary patterns or physical activity levels, potentially confounding the true effects of LPT. These two limitations may impact the internal validity and generalizability of the findings. Future research should incorporate a control group and meticulously monitor lifestyle factors to more accurately evaluate the independent effects of LPT on metabolic health.

### 4.3. Clinical Implications and Future Directions

This study demonstrates that LPT as a natural, non-pharmacological intervention, has a broad clinical application prospect in the prevention and management of metabolic diseases. Future studies should focus on addressing the current limitations of the research, such as increasing the sample size, extending the intervention period, incorporating a control group, and systematically monitoring participants’ lifestyle factors, in order to more accurately assess the independent health effects of LPT.

## 5. Conclusions

LPT significantly regulates blood pressure, lipid and glucose metabolism, reduces body weight, and improves body composition parameters, while actively modulating gut microbiota. These effects do not show significant differences across LPT aged for different durations, challenging the traditional belief that longer-aged LPT possesses superior health advantages. The study results indicate that LPT, as a functional beverage, has the potential to promote metabolic health, warranting further research into its mechanisms and long-term impacts.

## Figures and Tables

**Figure 1 foods-14-00866-f001:**
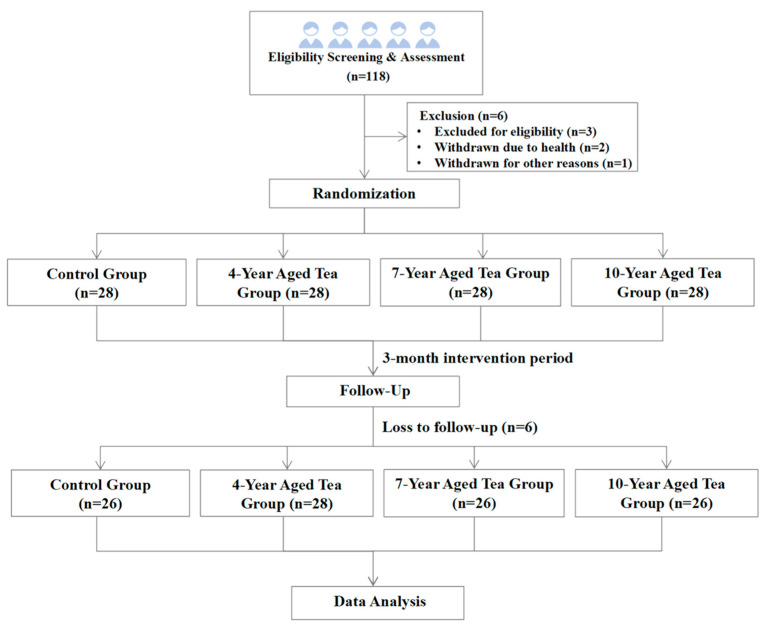
Study flowchart from participant recruitment to data analysis.

**Figure 2 foods-14-00866-f002:**
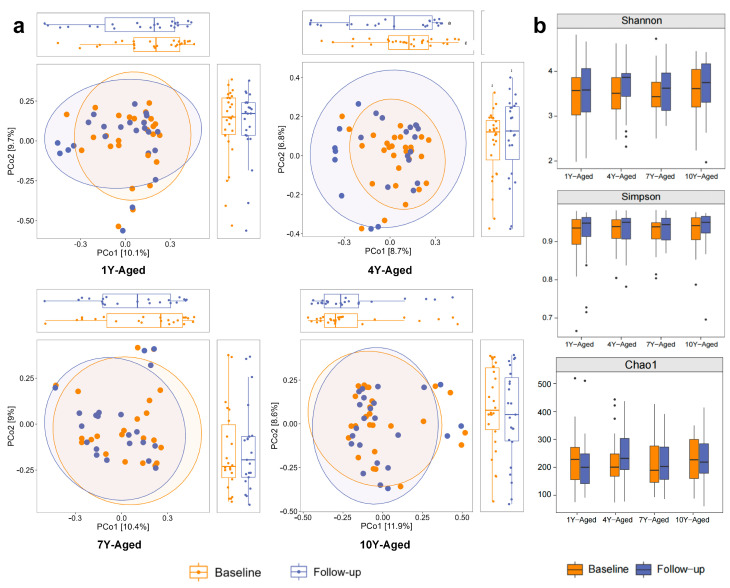
Principal coordinate analysis (PCoA) and alpha diversity of gut microbiota across four LPT groups with different years of aging. (**a**) PCoA plots showed the differences in gut microbiota structure before and after intervention in the 1-year, 4-year, 7-year, and 10-year-aged groups. (**b**) Box plots of Shannon, Simpson, and Chao1 indices measuring alpha diversity in the four groups.

**Figure 3 foods-14-00866-f003:**
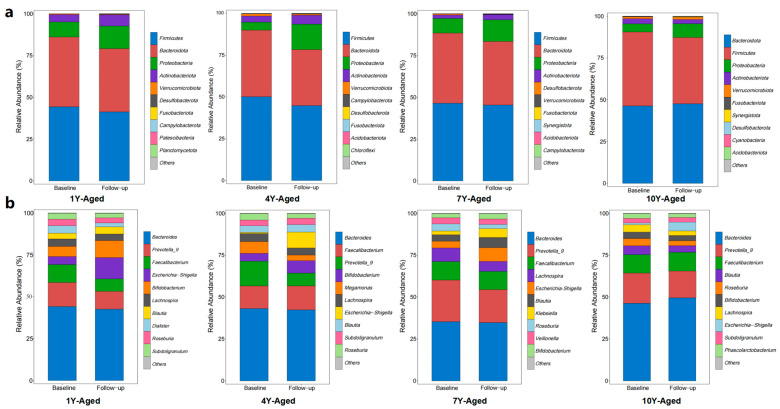
Changes in the relative abundance of gut microbiota at the phylum (**a**) and genus (**b**) levels before and after intervention with LPT of different years of aging.

**Figure 4 foods-14-00866-f004:**
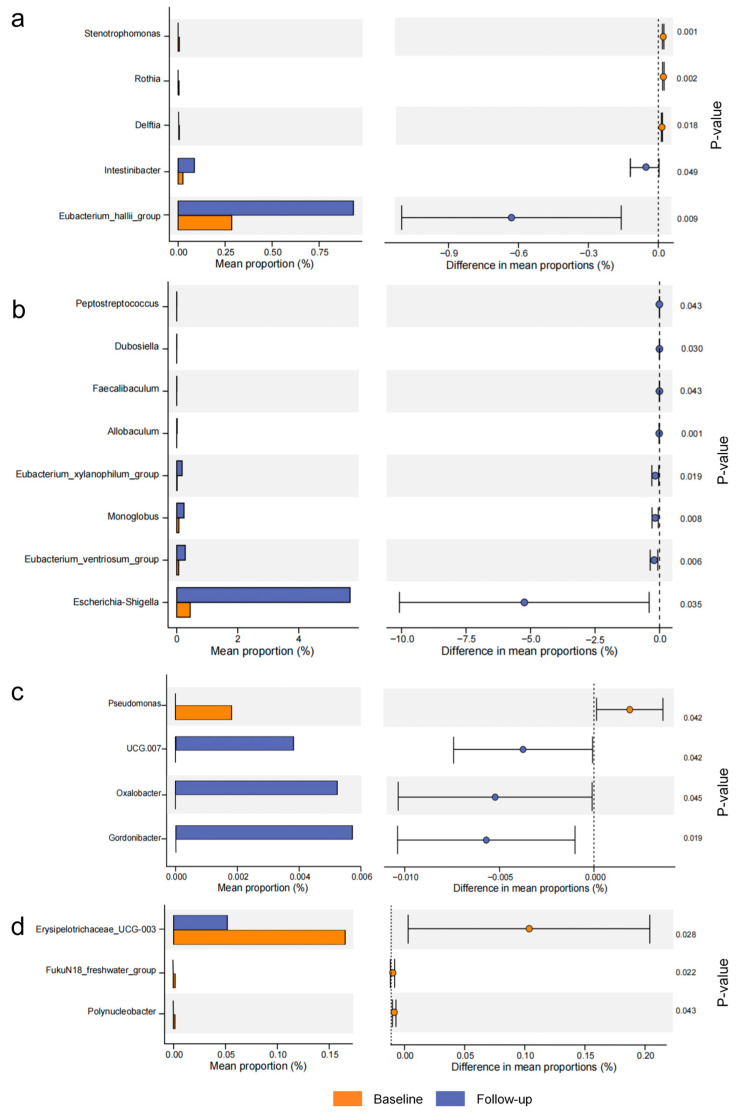
Analysis of the gut bacteria at the genus level in LPT groups with different years of aging (only showing results with statistical significance). (**a**) 1-year-aged group; (**b**) 4-year-aged group; (**c**) 7-year-aged group; (**d**) 10-year-aged group.

**Table 1 foods-14-00866-t001:** Baseline characteristics of the total population and different aged LPT intervention groups ^a^.

Characteristic	Total Population Group (*n* = 106)	1-Year-Aged Group (*n* = 26)	4-Year-Aged Group (*n* = 28)	7-Year-Aged Group (*n* = 26)	10-Year-Aged Group (*n* = 26)	*p*
Age (years), mean ± SD	50.9 ± 17.4	51.3 ± 17.9	49.5 ± 17.7	51.2 ± 17.2	51.8 ± 17.5	0.968
Gender, *n* (%)						0.974
Male	28 (26.4)	7 (26.9)	8 (28.6)	7 (26.9)	6 (23.1)	
Female	78 (73.6)	19 (73.1)	20 (71.4)	19 (73.1)	20 (76.9)	
Marital Status, *n* (%)						0.790
Single	17 (16.0)	4 (15.4)	6 (21.4)	4 (15.4)	3 (11.5)	
Married/cohabiting	79 (74.5)	21 (80.8)	20 (71.4)	19 (73.1)	19 (73.1)	
Divorced/widowed	10 (9.4)	1 (3.8)	2 (7.2)	3 (11.5)	4 (15.4)	
Educational Level, *n* (%)						0.917
Junior high school and below	8 (7.5)	2 (7.7)	2 (7.1)	2 (7.7)	2 (7.7)	
High school	16 (15.1)	3 (11.5)	3 (10.7)	4 (15.4)	6 (23.1)	
Bachelor’s degree and above	82 (77.4)	21 (80.8)	23 (82.1)	20 (76.9)	18 (69.2)	
Average Monthly Income, *n* (%)						0.528
<3000 CNY	18 (17.0)	4 (15.4)	4 (14.2)	5 (19.2)	5 (19.2)	
3000~8000 CNY	42 (39.6)	11 (42.3)	12 (42.9)	6 (23.1)	13 (50.0)	
≥8000 CNY	46 (43.4)	11 (42.3)	12 (42.9)	15 (57.7)	8 (30.8)	
Smoking, *n* (%)						0.021
Never smoked	99 (93.4)	23 (88.5)	24 (85.7)	26 (100.0)	26 (100.0.)	
Former smoker	4 (3.8)	3 (11.5)	1 (3.6)	0 (0.0)	0 (0.0)	
Current smoker	3 (2.8)	0 (0.0)	3 (10.7)	0 (0.0)	0 (0.0)	
Alcohol Consumption, *n* (%)						0.273
Never drinks	94 (88.7)	23 (88.5)	22 (78.5)	24 (92.3)	25 (96.2)	
Former drinker	9 (8.5)	3 (11.5)	5 (17.9)	1 (3.9)	0 (0.0)	
Current drinker	3 (2.8)	0 (0.0)	1 (3.6)	1 (3.8)	1 (3.8)	
Frequent Tea Drinking, *n* (%)						0.744
Yes	49 (46.2)	12 (46.2)	15 (53.6)	12 (46.2)	10 (38.5)	
No	57 (53.8)	14 (53.8)	13 (46.4)	14 (53.8)	16 (61.5)	
Medical History, *n* (%)						
T2DM	84 (79.2)	22 (84.6)	23 (82.1)	17 (65.4)	22 (84.6)	0.100
Hypertension	71 (67.0)	16 (61.5)	19 (67.9)	18 (69.2)	18 (69.2)	0.756
Dyslipidemia	65 (61.3)	17 (65.4)	14 (50)	15 (57.7)	19 (73.1)	0.526
Heart disease	98 (92.5)	22 (84.6)	26 (92.9)	25 (96.2)	25 (96.2)	0.526

^a^ Continuous variables are presented as mean ± SD; categorical variables are shown as frequencies (n) and percentages (%); SD, standard deviation; CNY, Chinese Yuan; T2DM, Type 2 diabetes.

**Table 2 foods-14-00866-t002:** The intervention effects of LPT with different years of aging on metabolic parameters ^a^.

Metabolic Parameters ^b^	1-Year-Aged Group (*n* = 26)		4-Year-Aged Group (*n* = 28)		7-Year-Aged Group (*n* = 26)		10-Year-Aged Group (*n* = 26)	
	Baseline	Follow-Up	Baseline	Follow-Up	Baseline	Follow-Up	Baseline	Follow-Up
Metabolic parameters								
SBP, mmHg	129.84 ± 22.35	122.92 ± 21.47 *	130.70 ± 14.48	121.37 ± 17.15 *	131.38 ± 13.55	115.30 ± 23.93 *	131.32 ± 16.08	121.16 ± 15.16 **
DBP, mmHg	76.28 ± 12.79	73.08 ± 12.67 *	78.41 ± 8.54	73.56 ± 11.59 *	78.54 ± 10.75	74.08 ± 15.91	81.72 ± 14.56	72.00 ± 10.94 **
TC, mmol/L	5.10 ± 1.08	4.87 ± 0.88 *	5.01 ± 1.12	4.83 ± 1.10	5.19 ± 0.91	5.38 ± 0.85	5.26 ± 0.79	4.96 ± 0.86
TG, mmol/L	1.26 ± 0.82	1.35 ± 0.88	1.28 ± 1.09	1.08 ± 0.51	1.63 ± 0.87	1.66 ± 1.16	1.73 ± 1.86	1.63 ± 1.61
HDL-C, mmol/L	1.34 ± 0.36	1.36 ± 0.31	1.30 ± 0.23	1.36 ± 0.21	1.29 ± 0.29	1.37 ± 0.25 *	1.34 ± 0.22	1.37 ± 0.18
LDL-C, mmol/L	2.91 ± 0.68	2.63 ± 0.55 *	2.90 ± 0.83	2.65 ± 0.76 *	3.08 ± 0.64	2.94 ± 0.50	2.95 ± 0.52	2.63 ± 0.55 *
APOA1, g/L	1.33 ± 0.23	1.25 ± 0.21 *	1.30 ± 0.16	1.22 ± 0.14 *	1.31 ± 0.16	1.24 ± 0.14 *	1.36 ± 0.15	1.25 ± 0.12 **
APOB, g/L	0.85 ± 0.17	0.82 ± 0.14	0.84 ± 0.21	0.83 ± 0.24	0.88 ± 0.16	0.88 ± 0.11	0.83 ± 0.15	0.82 ± 0.14
HbA1c, %	5.59 ± 1.00	5.68 ± 0.99 *	5.60 ± 0.68	5.63 ± 0.68	5.87 ± 1.21	5.92 ± 1.12	5.79 ± 1.29	5.87 ± 1.37
FBG, mmol/L	5.98 ± 1.87	5.88 ± 2.09	5.75 ± 1.19	5.50 ± 1.07 *	8.38 ± 11.56	6.05 ± 1.69	6.05 ± 2.47	5.90 ± 1.97
INS, pmol/L	75.93 ± 33.55	71.70 ± 44.12	73.31 ± 31.06	62.21 ± 28.87	72.68 ± 35.16	67.78 ± 35.83	83.19 ± 50.47	74.28 ± 46.00
Body composition								
Weight, kg	69.70 ± 13.81	68.21 ± 13.19 **	69.00 ± 11.96	66.70 ± 10.77 **	67.36 ± 10.62	65.33 ± 10.92 *	67.06 ± 11.3	65.57 ± 11.90 **
BFM, kg	21.78 ± 5.24	20.28 ± 4.77 **	21.88 ± 5.64	19.54 ± 5.59 **	21.13 ± 4.66	19.15 ± 4.44 **	22.09 ± 6.08	20.12 ± 5.43 **
LBM, kg	46.34 ± 9.17	46.92 ± 9.77 *	47.12 ± 9.02	47.63 ± 9.44	46.61 ± 8.07	46.83 ± 8.59	46.33 ± 9.93	47.01 ± 11.38
MM, kg	42.48 ± 8.49	43.09 ± 9.09 *	42.08 ± 11.06	43.43 ± 9.03	42.76 ± 7.52	43.06 ± 8.00	42.46 ± 9.21	43.18 ± 10.61
BMI, kg/m^2^	25.90 ± 3.24	25.44 ± 3.12 *	26.22 ± 2.99	25.38 ± 2.76 **	25.19 ± 2.65	24.47 ± 2.80 *	25.47 ± 3.17	24.93 ± 3.29 *
WC, cm	84.51 ± 8.43	82.60 ± 7.33 **	85.55 ± 8.21	82.04 ± 6.71 **	84.66 ± 7.91	82.20 ± 7.74 **	84.60 ± 9.65	79.37 ± 17.19
BFP, %	31.54 ± 4.98	30.21 ± 4.56 *	31.62 ± 6.37	29.10 ± 7.07 **	31.18 ± 5.16	29.02 ± 4.97 **	32.20 ± 5.62	30.07 ± 5.66 **
VFA, cm^2^	99.13 ± 40.25	86.52 ± 33.63 **	100.85 ± 36.11	80.56 ± 31.05 **	93.96 ± 29.61	79.29 ± 27.78 **	101.60 ± 39.97	85.00 ± 32.52 **

** *p* < 0.001, * *p* < 0.05. ^a^ Values are expressed as mean ± standard. ^b^ Abbreviations: SBP: systolic blood pressure; DBP: diastolic blood pressure; TC: total cholesterol; TG: triglycerides; HDL-C: high-density lipoprotein cholesterol; LDL-C: low-density lipoprotein cholesterol; APOA1: apolipoprotein A1; APOB: apolipoprotein B; HbA1c: hemoglobin A1c; FBG: fasting blood glucose; INS: insulin; BFM: body fat mass; LBM: lean body mass; MM: muscle mass; BMI: body mass index; WC: waist circumference; BFP: body fat percentage; VFA: visceral fat area.

## Data Availability

The data presented in this study are available on request from the corresponding author due to privacy concerns.

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
