# Peer review of "Effects of Liupao Tea with Different Years of Aging on Glycolipid Metabolism, Body Composition, and Gut Microbiota in Adults with Obesity or Overweight: A Randomized, Double-Blind Study"

_foods, 2025, doi:10.3390/foods14050866_

Round 1
Reviewer 1 Report
Comments and Suggestions for Authors
Liupao tea (LPT), a type of dark tea produced in Guangxi, China, is renowned for its unique fermentation and aging processes, which give it distinctive chemical components and bioactivities. Compared to green tea, LPT contains higher alkaloids, fatty acids, and relatively more significant amounts of ellagic acid, hypoxanthine, and theobromine.
LPT has been shown to significantly regulate blood pressure, lipid levels, and glucose metabolism. It also contributes to weight reduction and improves body composition while actively modulating gut microbiota. Interestingly, the effects of LPT do not vary significantly based on its aging duration, challenging the traditional belief that longer-aged LPT offers superior health benefits. The findings suggest that LPT can serve as a functional beverage that promotes metabolic health, indicating a need for further research into its mechanisms and long-term effects.
Strength of the manuscript: The degree of novelty, namely, the first-time examination of the influence of LPT aging on the biochemical and physiological characteristics of the experiment participants. The manuscript is clearly and comprehensibly written, and the experimental data are presented graphically.
The weakness of the manuscript is the absence of chemical characterization of LPT at different ages.
Suggested correction:
1. In the introductory part, briefly write about the conditions under which the examined tea is stored during aging, temperature, humidity, airflow, etc.
2. Whether the LPT originates from the same or a different geographical region.
3. If the authors can, it would be good to do chemical analyses (HPLC) for the main fractions of the tested teas (HPLC) or record IR spectra to analyze whether the functional groups change during aging. If the company that produces the tea has some chemical results, the same can be provided.
Author Response
Reviewer 1
Dear Reviewer,
First of all, we would like to express our sincere gratitude for your careful review of our manuscript and for providing constructive feedback. We have carefully revised the manuscript based on your suggestions. Below, we provide a point-by-point response to your feedback, hoping that these revisions will better highlight the value and significance of our research.
Comments 1:The degree of novelty, namely, the first-time examination of the influence of LPT aging on the biochemical and physiological characteristics of the experiment participants. The manuscript is clearly and comprehensibly written, and the experimental data are presented graphically.
Response 1: Thank you for your kind comments.
Comments 2:The weakness of the manuscript is the absence of chemical characterization of LPT at different ages.
Response 2: Thank you for your valuable feedback. We have added the analysis data of the main water-soluble component contents in LPT at different aging stages in the supplementary materials (Table S1). Additionally, we have included a discussion of these analysis results in the “Discussion” sections. (Page14-15 Line489-499)
Comments 3: In the introductory part, briefly write about the conditions under which the examined tea is stored during aging, temperature, humidity, airflow, etc.
Response 3: Thank you for your helpful suggestions. We have made detailed additions in the “Materials and Methods” section of the manuscript. Specifically, during the aging process, the samples were stored in a temperature-controlled and humidity-controlled warehouse with a constant temperature of 20-25°C and a relative humidity of 60-65%, maintaining appropriate air circulation and without direct sunlight exposure. (Page3 Line127-130)
Comments 4: Whether the LPT originates from the same or a different geographical region.
Response 4: Thank you for your valuable suggestions. The LPT samples used in this experiment from four different years (1 year, 4 years, 7 years, and 10 years) all originate from Zhaoping County, Guangxi Zhuang Autonomous Region, ensuring consistency in geographical origin. We have made this addition in the “Materials and Methods” section. (Page3 Line126-127)
Comments 5: If the authors can, it would be good to do chemical analyses (HPLC) for the main fractions of the tested teas (HPLC) or record IR spectra to analyze whether the functional groups change during aging. If the company that produces the tea has some chemical results, the same can be provided.
Response 5: Thank you for your valuable comments. In response to your suggestion regarding chemical analyses, we have supplemented the results of the main water-soluble component content in LPT of different aging years (e.g., tea polyphenols, caffeine, and flavonoids) in the supplementary materials (TableS1), in order to more comprehensively demonstrate the changes in the chemical composition of the tea leaves during the aging process. Additionally, in the “Discussion” section, we have further supplemented the potential impact of differences in chemical composition in LPT of different aging years on their health effects. Thank you once again for your professional suggestions. (Page14 Line489-499)
Reviewer 2 Report
Comments and Suggestions for Authors
The submitted publication concerning the impact of Liupao tea of different aging periods on glycolipid metabolism, body composition, and gut microbiota in overweight and obese individuals presents interesting results but contains several elements requiring improvement and supplementation.
First, attention should be given to the structure of the work. The introduction is relatively short and does not fully cover the topic. The authors present basic data on the global problem of obesity and the potential health benefits of Liupao tea consumption; however, a broader literature review on other fermented teas and their impact on metabolic health is missing. Expanding this section could enhance the study's cognitive value.
From a methodological perspective, the study is based on a solid approach involving a randomized, double-blind design. The description of participant recruitment and inclusion criteria is clear and precise. Nevertheless, a significant limitation is the lack of a placebo-controlled group, which makes it difficult to attribute the observed effects of the intervention solely to Liupao tea consumption. This study design makes it impossible to rule out the influence of time or other external factors on the results.
The findings concerning reductions in blood pressure, improved lipid profiles, and weight loss are interesting, but no statistically significant differences were found between groups consuming teas aged for different periods. This challenges the traditional belief that longer-aged tea has superior health properties. The authors suggest that the aging of tea does not significantly influence its metabolic effect, but the lack of detailed chemical composition analysis of teas in the different groups limits the ability to draw definitive conclusions.
The analysis of gut microbiota is an essential part of the work. The authors conducted studies on alpha and beta diversity and identified changes in the dominance of certain bacterial genera. However, the absence of functional microbiome analysis limits the understanding of the actual impact of Liupao tea on microbiome functions. It is also worth noting that the changes in microbiota were not significant between the groups consuming teas of different aging periods.
In terms of presenting results, the PCoA plots and microbiota diversity charts are clear, but some descriptions require clarification. Adding charts depicting differences between groups in metabolic parameters would facilitate the visualization of significant intervention effects.
The bibliography is largely based on current sources, although it would be worthwhile to consider including the latest publications from 2024 to strengthen the literature context of the work. The formatting is consistent with the journal's requirements, but attention should be paid to inconsistent citation styles in tables and charts.
In summary, the study makes a valuable contribution to research on the potential impact of Liupao tea on metabolic health, but the lack of a control group, limitations in microbiota analysis, and incomplete chemical composition analysis of teas are significant shortcomings. Expanding the introduction and developing the discussion on limitations would significantly improve the quality of this publication.
Questions for the authors:
- Why was a placebo-controlled group not included?
- Were additional functional analyses of the microbiota (e.g., metagenomics) conducted?
- What factors might have influenced the lack of statistically significant differences between the tea aging groups?
- Were changes in participants' diet and physical activity levels monitored during the study?
- Are there any plans for long-term studies to assess the durability of the effects of Liupao tea consumption?
Suggestions for Study Supplementation:
- Placebo Control: It is recommended to introduce a placebo control group in future studies to better assess the specific effects of Liupao tea.
- Functional Microbiota Analysis: Conducting metagenomic studies would allow for a better understanding of changes in gut microbiota functions.
- Chemical Composition of Tea: Detailed chemical analysis of the tea (including polyphenols, catechins, and metabolites) across different aging groups would help explain the lack of differences in health effects.
- Longer Observation Period: Conducting long-term studies would enable the assessment of the durability of health effects.
- Lifestyle Monitoring: Systematic recording of changes in participants' diet and physical activity would help exclude potential confounding factors.
The results in the publication are presented in a generally clear manner. Tables and figures, such as PCoA plots and microbiota diversity charts, effectively visualize key findings. However, additional visual elements, particularly graphs comparing changes in metabolic parameters between tea-aging groups, could improve the clarity and accessibility of the data. Descriptions accompanying the figures sometimes lack sufficient detail or context, which may make interpretation difficult for readers unfamiliar with specific methods or statistical analyses.
The conclusions are partially supported by the results. The authors assert that Liupao tea positively affects blood pressure, lipid profiles, body composition, and gut microbiota, and that its health benefits are consistent regardless of aging duration. While the results show significant improvements in metabolic parameters across all intervention groups, the lack of statistically significant differences between tea-aging groups undermines the strength of this assertion. Additionally, the absence of a placebo group raises questions about the attribution of observed effects solely to tea consumption. The conclusions could be more robust by acknowledging these limitations and discussing alternative interpretations of the data.
Author Response
Point-by-Point Response to Reviewer Comments
Reviewer 2
Dear Reviewer,
First of all, we would like to express our sincere gratitude for your careful review of our manuscript and for providing constructive feedback. We have carefully revised the manuscript based on your suggestions. Below, we provide a point-by-point response to your feedback, hoping that these revisions will better highlight the value and significance of our research.
Comments 1: First, attention should be given to the structure of the work. The introduction is relatively short and does not fully cover the topic. The authors present basic data on the global problem of obesity and the potential health benefits of Liupao tea consumption; however, a broader literature review on other fermented teas and their impact on metabolic health is missing. Expanding this section could enhance the study's cognitive value.
Response 1: Thank you for your valuable suggestions. To better enhance the cognitive value of our research, we have added information on several other major fermented teas and their potential impacts on metabolic health in the introduction section. This addition aims to help readers better understand the significance of our study. (Page2 Line46-62)
Comments 2: From a methodological perspective, the study is based on a solid approach involving a randomized, double-blind design. The description of participant recruitment and inclusion criteria is clear and precise. Nevertheless, a significant limitation is the lack of a placebo-controlled group, which makes it difficult to attribute the observed effects of the intervention solely to Liupao tea consumption. This study design makes it impossible to rule out the influence of time or other external factors on the results.
Response 2: Thank you for your valuable comments. We acknowledge that the lack of a placebo-controlled group may be a limitation of the study. In this research, due to practical and ethical considerations, we were unable to include a placebo-controlled group. However, we have made every effort to ensure the scientific rigor and reliability of the study, including the use of randomization and double-blind design to minimize potential biases. We will further discuss this limitation in the discussion section and suggest possible improvements for future studies, such as incorporating a placebo-controlled group, to better control for the influence of time or other external factors on the results. (Page15 Line504-517)
Comments 3: The findings concerning reductions in blood pressure, improved lipid profiles, and weight loss are interesting, but no statistically significant differences were found between groups consuming teas aged for different periods. This challenges the traditional belief that longer-aged tea has superior health properties. The authors suggest that the aging of tea does not significantly influence its metabolic effect, but the lack of detailed chemical composition analysis of teas in the different groups limits the ability to draw definitive conclusions.
Response 3: Thank you for your valuable comments. As the reviewer mentioned, the lack of detailed chemical composition analysis of the teas in different groups indeed limits our ability to draw definitive conclusions. To address this, we have provided the main water-soluble components in LPT from different aging years in the supplementary materials (Table S1). Additionally, we further discuss the potential impact of these chemical composition changes on metabolic effects in the “Discussion” section and highlight the need for larger sample sizes and longer intervention periods in future studies to validate these findings. (Page14 Line489-499)
Comments 4: The analysis of gut microbiota is an essential part of the work. The authors conducted studies on alpha and beta diversity and identified changes in the dominance of certain bacterial genera. However, the absence of functional microbiome analysis limits the understanding of the actual impact of Liupao tea on microbiome functions. It is also worth noting that the changes in microbiota were not significant between the groups consuming teas of different aging periods.
Response 4: Thank you for your kind suggestions. To further explore the potential impact of Liupao tea with different aging years on microbial functions, we employed the PICRUSt2 functional prediction method and analyzed the microbial community functions of Liupao tea aged for 1, 4, 7, and 10 years based on the KO database. Specifically, we selected the top ten functional features by relative abundance in each group and performed Wilcoxon signed-rank test to screen for significantly different functions (P < 0.05). However, after analysis, we found that none of the top ten functional features in the four different aged LPT groups showed significant differences in the Wilcoxon signed-rank test. The results of the functional analysis are presented in the supplementary materials (Table S4).
Additionally, regarding the issue that the changes in microbiota were not significant between the groups consuming teas of different aging periods, we also noted this point. The possible reasons may include limitations in sample size, individual differences, and whether the aging periods were sufficiently distinct to differentiate microbial changes. We plan to increase the sample size and optimize the experimental design (such as extending the aging periods or adding more time points) in future studies.
In the "Methods" section, we supplemented the specific steps and analysis methods of PICRUSt2 functional prediction. (Page6 Line267-272)
In the "Results" section, we supplemented the top ten functional pathways for each aged LPT group, their relative abundance, and the results of the Wilcoxon signed-rank test. (Page10 Line372-374)
In the "Discussion" section, we further explored the reasons for these results, understanding that this may be related to a variety of factors, such as sample size and the accuracy of functional prediction. In future studies, we plan to further optimize the analysis methods, increase the sample size, or combine other functional analysis tools to more comprehensively evaluate the impact of Liupao tea with different aging years on microbial community functions. (Page14 Line474-479)
Comments 5: In terms of presenting results, the PCoA plots and microbiota diversity charts are clear, but some descriptions require clarification. Adding charts depicting differences between groups in metabolic parameters would facilitate the visualization of significant intervention effects.
Response 5: Thank you for your valuable comments. Regarding the description of the microbiota results, we have further clarified the relevant content to ensure that readers can more easily understand it. As for the reviewer's suggestion to add charts depicting differences in metabolic parameters between groups, we fully agree with this suggestion. In the supplementary materials, we have used box plots to visualize the differences in the effects of Liupao tea with different aging years on improving metabolic parameters and body composition, to more clearly present the intervention effects. Thank you again for your suggestions. We believe these revisions will help improve the readability of the manuscript and the presentation of the data.
Comments 6: The bibliography is largely based on current sources, although it would be worthwhile to consider including the latest publications from 2024 to strengthen the literature context of the work. The formatting is consistent with the journal's requirements, but attention should be paid to inconsistent citation styles in tables and charts.
Response 6: Thank you sincerely for your valuable comments. We have incorporated the latest publications from 2024 into the manuscript to strengthen the literature context. Additionally, we have reviewed the formatting of citations in tables and charts to ensure consistency throughout the manuscript, as per the journal’s guidelines.
Comments 7: In summary, the study makes a valuable contribution to research on the potential impact of Liupao tea on metabolic health, but the lack of a control group, limitations in microbiota analysis, and incomplete chemical composition analysis of teas are significant shortcomings. Expanding the introduction and developing the discussion on limitations would significantly improve the quality of this publication.
Response 7: Thank you sincerely for your constructive suggestions. We truly appreciate the time and effort you have taken to evaluate our research. In response to your feedback, we have made the following revisions and improvements:
Regarding the lack of a control group: While practical and ethical considerations prevented us from including a control group in the current study, we have provided a detailed discussion of this limitation in the discussion section. We plan to include a placebo control group in future studies to better validate the effects of Liupao tea on health outcomes. (Page15 Line504-517)
Regarding the limitations in microbiota analysis: We acknowledge the importance of your comments on the limitations of microbiota analysis. To address this, we have added functional prediction results using PICRUST2 (as previously described). Furthermore, we plan to employ metagenomic sequencing in our next study to provide a more comprehensive analysis of gut microbiota composition and function. (Page14 Line474-479)
Regarding the incompleteness of tea chemical composition analysis: We have supplemented the analysis by including detailed quantification of the major water-soluble components of Liupao tea samples with different aging durations in the supplementary materials (see Table S1). These data include quantitative information on key compounds such as tea polyphenols and catechins, as well as their changing trends. We have also further discussed the impact of these compositional changes on metabolic health in the discussion section. (Page14 Line489-499)
Expanding the introduction and developing the discussion on limitations: To better reflect the background and limitations of the study, we have expanded the introduction to include a more comprehensive review of the literature on fermented tea and its effects on metabolic health. Additionally, we have elaborated on the limitations of the current study and proposed future research directions in the discussion section. (Page2 Line46-62, Page15 Line518-524)
We believe these revisions will significantly enhance the overall quality of the manuscript. Once again, we are deeply grateful for your valuable suggestions and look forward to your further feedback.
Comments 8: Why was a placebo-controlled group not included?
Response 8: Thank you for your constructive questions. In this study, due to practical and ethical considerations, we were unable to include a placebo-controlled group. Specifically, as Liupao tea is a traditional functional tea, its health effects are the core focus of our research, and using a placebo could potentially affect participants' informed consent and the ethical nature of the study. Therefore, we have implemented measures such as randomization and a double-blind design to minimize potential biases. Additionally, we have discussed this limitation in detail in the 'Discussion' section and suggested that future studies could include a placebo-controlled group to better assess the intervention effects of Liupao tea. (Page15 Line504-517)
Comments 9: Were additional functional analyses of the microbiota (e.g., metagenomics) conducted?
Response 9: Thank you for your kind comments. We used 16S rRNA sequencing to analyze the composition of the gut microbiota. This method is highly efficient and cost-effective in assessing microbial diversity, relative abundance, and their changes. However, we also recognize the limitations of 16S rRNA sequencing in functional analysis. Due to resource and technical constraints, we did not conduct additional functional analyses (such as metagenomic sequencing). To partially address this limitation, we used the PICRUSt2 tool to predict functional potential based on 16S rRNA data, aiming to infer the functional potential of the microbiota. We agree that more in-depth functional analyses, such as metagenomic sequencing, are crucial for a comprehensive understanding of gut microbiota functions and metabolic pathways. Therefore, in future extended studies, we plan to use metagenomic sequencing technology to obtain more comprehensive functional information on the microbiota and further validate the findings of the current study.
Comments 10: What factors might have influenced the lack of statistically significant differences between the tea aging groups?
Response 10: Thank you for your question. Although traditional beliefs suggest that longer-aged Liupao Tea (LPT) possesses superior quality, our study found no significant differences in the effects of LPT aged for different durations on improving blood pressure, blood lipids, blood glucose, body weight, or body composition parameters. These results may be influenced by the following factors:
First, as the aging time increases, the major bioactive components of LPT undergo significant changes. The catechin content decreases gradually from 12.5% to 11.0%, and theaflavins decrease from 2.9% to 2.5%, while flavonoid compounds show a trend of first increasing and then decreasing, from 2.4% to 3.2%, and then dropping to 2.2%. This non-linear variation may result in a peak in health effects at a certain aging time, after which further extension of the aging period does not lead to better health outcomes and may even slightly decrease due to the reduction of bioactive components.
Second, compared to animal models, the human metabolic system is influenced by a variety of factors, including individual differences, diet, lifestyle, and genetic background. These complex factors may mask the small differences in health effects of LPT aged for different durations.
Additionally, our study may have some limitations in experimental design, such as sample size, intervention duration, and the selection of health indicators, which may not be sufficient to detect the subtle differences in health effects caused by aging time. Larger-scale and longer-term studies may be necessary to fully capture these potential differences.
Lastly, LPT contains multiple bioactive components that may interact synergistically or antagonistically in the body. For example, catechins and flavonoid compounds may work together in antioxidant activity and metabolic regulation, while other components, such as soluble dietary fiber, may regulate metabolic health through gut microbiota. These complex interactions may lead to similar health effects across different aging times, even if the content of individual components varies.
In summary, although aging time significantly affects the composition and flavor of LPT, its impact on human metabolic health is likely more complex and involves the interaction of multiple factors. We plan to further explore the relationships between these factors in future research, including the interactions between components and their long-term effects on human health, to provide a more comprehensive evaluation of the health benefits of LPT. We have added a discussion of this in the “Discussion” section to better explain these results. (Page14 Line489-499)
Comments 11: Were changes in participants' diet and physical activity levels monitored during the study?
Response 11: Thank you for your valuable suggestions. In response to your question, "Were changes in participants' diet and physical activity levels monitored during the study?" we did not specifically record the dietary and physical activity levels of the participants during the study. However, we explicitly advised the participants to maintain their usual diet, exercise, and sleep routines without making any significant lifestyle changes during the study period. Given that the intervention lasted for 90 days, this relatively short timeframe may have limited our ability to comprehensively monitor changes in diet and physical activity.
Nevertheless, we acknowledge the potential impact of diet and physical activity on the study outcomes. Therefore, we attempted to minimize the influence of these variables by advising participants to adhere to their regular habits. We understand that this approach may not completely eliminate all potential confounding factors, but it was a reasonable measure under the current research conditions.
In future studies, we plan to improve our research design by implementing more detailed records and analyses of participants' diet and physical activity. This will allow us to more fully assess the impact of these factors on the study outcomes. Thank you for your valuable suggestions. We will supplement this explanation in the discussion section of the paper to better clarify our research methods and results. (Page15 Line521-524)
Comments 12: Are there any plans for long-term studies to assess the durability of the effects of Liupao tea consumption?
Response 12: Thank you for your valuable suggestions regarding our study. We fully agree with the importance of long-term research in assessing the sustained effects of Liupao tea consumption. In the current study, we conducted only a 90-day intervention. Although the results showed significant effects of Liupao tea with different aging periods on improving blood pressure, blood lipids, blood glucose, weight, and body composition, we recognize that this is merely a short-term observation and does not comprehensively demonstrate the long-term health impacts of Liupao tea. Therefore, we plan to conduct long-term follow-up and evaluation in future studies to more thoroughly understand the lasting effects of Liupao tea consumption on human health.
In these future long-term studies, we plan to adopt more comprehensive assessment metrics, including but not limited to metabolomics analysis, to gain deeper insights into the mechanisms of Liupao tea and its long-term metabolic impacts on the human body. Additionally, we intend to explore individual differences among various populations, such as age, gender, and genetic background, to provide a more robust scientific basis for the health benefits of Liupao tea. (Page15 Line521-524)
Comments 13: Placebo Control: It is recommended to introduce a placebo control group in future studies to better assess the specific effects of Liupao tea.
Response 13: Thank you for your valuable suggestion regarding our research. We completely agree on the importance of introducing a placebo control group to better assess the specific effects of Liupao tea. In the current study, due to study design and resource limitations, we were unable to include a placebo control group. However, we will actively consider this suggestion in future studies and incorporate a placebo control group to more rigorously validate the health effects of Liupao tea. Thank you again for your input, and we will continue to refine our study design in subsequent research to provide more reliable scientific evidence. (Page15 Line521-524)
Comments 14: Functional Microbiota Analysis: Conducting metagenomic studies would allow for a better understanding of changes in gut microbiota functions.
Response 14: Thank you for your suggestion regarding the use of metagenomics to better understand changes in gut microbiota functions. As mentioned earlier, due to resource and technical constraints, we employed 16S rRNA sequencing to analyze the composition of the gut microbiota. In future studies, we plan to incorporate metagenomic analysis to more comprehensively explore the effects of Liupao tea on gut microbiota functions. This will help deepen our understanding of the mechanisms by which Liupao tea influences health outcomes. (Page14 Line474-479)
Comments 15: Chemical Composition of Tea: Detailed chemical analysis of the tea (including polyphenols, catechins, and metabolites) across different aging groups would help explain the lack of differences in health effects.
Response 15: Thank you for your valuable suggestion. We completely agree that detailed chemical composition analysis of tea would help better explain the differences in health effects across different aging groups. We have provided the main water-soluble components of Liupao tea from different aging years in the supplementary materials (Table S1). In future studies, we plan to conduct more detailed chemical analyses to help clarify how changes in tea components influence health effects and further explain the observations in our current study. (Page14 Line489-499)
Comments 16: Longer Observation Period: Conducting long-term studies would enable the assessment of the durability of health effects.
Response 16: Thank you for your kind suggestion. We completely agree that conducting long-term studies would better assess the health effects of Liupao tea. While the current study used a 90-day intervention period, we recognize that the health effects of Liupao tea may require a longer duration to manifest or persist. Therefore, we plan to further optimize the study design and extend the observation period in future research to more comprehensively evaluate the long-term impact of Liupao tea on health. (Page15 Line521-524)
Comments 17: Lifestyle Monitoring: Systematic recording of changes in participants' diet and physical activity would help exclude potential confounding factors.
Response 17: Thank you for your suggestion. As we mentioned earlier, we did not specifically record changes in participants' diet and physical activity levels during the study. However, we advised participants to maintain their usual lifestyle habits. While this approach may not fully exclude potential confounding factors, it was the best possible strategy given the study design and timeframe. In future studies, we plan to implement more detailed monitoring of diet and physical activity to better assess their potential impact on the study outcomes. (Page15 Line510-517)
Comments 18: The results in the publication are presented in a generally clear manner. Tables and figures, such as PCoA plots and microbiota diversity charts, effectively visualize key findings. However, additional visual elements, particularly graphs comparing changes in metabolic parameters between tea-aging groups, could improve the clarity and accessibility of the data. Descriptions accompanying the figures sometimes lack sufficient detail or context, which may make interpretation difficult for readers unfamiliar with specific methods or statistical analyses.
Response 18: Thank you for your kind suggestion. We have presented box plots comparing the changes in metabolic parameters across different aging groups of Liupao tea in the supplementary materials to provide a more intuitive visualization of these differences. We also acknowledge the issue of insufficient detail and background information in the current figure descriptions. In the revised version, we have further expanded the figure legends to ensure that readers can better understand our results. We hope these revisions will enhance the readability and scientific value of the paper.
Comments 19: The conclusions are partially supported by the results. The authors assert that Liupao tea positively affects blood pressure, lipid profiles, body composition, and gut microbiota, and that its health benefits are consistent regardless of aging duration. While the results show significant improvements in metabolic parameters across all intervention groups, the lack of statistically significant differences between tea-aging groups undermines the strength of this assertion. Additionally, the absence of a placebo group raises questions about the attribution of observed effects solely to tea consumption. The conclusions could be more robust by acknowledging these limitations and discussing alternative interpretations of the data.
Response 19: Thank you for your valuable suggestions. We acknowledge that, while all intervention groups showed significant improvements in metabolic parameters, there was a lack of statistically significant differences between the different aging duration groups. We have provided further explanation of this result in the discussion section. Additionally, we plan to increase the sample size and extend the intervention period in future studies to more thoroughly investigate the impact of different aging durations on the health benefits of Liupao tea. Furthermore, we recognize that the absence of a placebo control group in the current study may raise questions about the attribution of observed effects solely to tea consumption. As mentioned previously, in future experimental designs, we will include a placebo control group to more rigorously validate these findings and rule out other potential factors.
Reviewer 3 Report
Comments and Suggestions for Authors
1) The adjective obese should be absolutely changed and substituted by “adults with obesity or overweight”, in all the manuscript, since the former is highly stigmatizing
2) I am not sure that authors can include highlights before the abstract according to MDPI publications guidelines
3) The design of the study should be clearly mentioned in the title
4) The statistical analysis tests are not should be necessarily mentioned in the abstract, this make the later more concise
5) The keywords should differs from those in the title, in order to increase the probability to find the article with a more literature searches
6) Following the aim in the introduction section, a hypothesis should be clearly formulated
7) Since authors has conducted multi-group comparison, the level of statistical analysis should be mandatory adjusted, such as using Bonferroni or other similar tests
8) Since 6 participants have dropped, how authors managed this? Authors should consider an intent-to-treat analysis including the people who dropped, rather than considering only the completers
9) The discussion section may benefit from a different arrangement, thus to be divided in 4 subsections as follows:
· The main findings of the study and their comparison with previously published papers
· The strengths and limitations of the study
· The clinical implications of the findings
· The new directions for future research
10) In the Authors’ contributions section please use the abbreviations of the authors
Author Response
Point-by-Point Response to Reviewer Comments
Reviewer 3
Dear Reviewer,
First of all, we would like to express our sincere gratitude for your careful review of our manuscript and for providing constructive feedback. We have carefully revised the manuscript based on your suggestions. Below, we provide a point-by-point response to your feedback, hoping that these revisions will better highlight the value and significance of our research.
Comments 1: The adjective obese should be absolutely changed and substituted by “adults with obesity or overweight”, in all the manuscript, since the former is highly stigmatizing.
Response 1: Thank you for pointing out the importance of using sensitive and respectful language in our manuscript. We fully agree with your suggestion to replace the term "obese" with "adults with obesity or overweight" throughout the document, as it is crucial to avoid stigmatizing language. We will make these changes in all relevant sections of the manuscript to ensure that our terminology is both accurate and respectful.
Comments 2: I am not sure that authors can include highlights before the abstract according to MDPI publications guidelines.
Response 2: Thank you for your valuable suggestions. We have reviewed the guidelines and removed the Highlights section as per your recommendation.
Comments 3: The design of the study should be clearly mentioned in the title.
Response 3: Thank you for your suggestion to clearly mention the study design in the title while adhering to the formatting guidelines of capitalizing only the first letter of the sentence. We have revised the title accordingly: Effects of liupao tea with different aged years on glycolipid metabolism, body composition, and gut microbiota in adults with obesity or overweight: a randomized, double-blind study. (Page1 Line1-4)
Comments 4: The statistical analysis tests are not should be necessarily mentioned in the abstract, this make the later more concise.
Response 4: Thank you for your kind suggestion. We have deleted the content related to statistical analysis tests in the abstract to make it more concise.
Comments 5: The keywords should differs from those in the title, in order to increase the probability to find the article with a more literature searches
Response 5: Thank you for your valuable suggestions. We have revised the keywords to better reflect the unique aspects of our study and to enhance discoverability. The updated keywords are as follows: Liupao tea; weight management; lipid profile; gut health; randomized trial. These new keywords aim to capture a wider range of topics related to our research while avoiding overlap with the title. We believe this will improve the visibility and accessibility of our article in relevant literature searches. We have already revised and supplemented the keywords in the abstract section. (Page1 Line29)
Comments 6: Following the aim in the introduction section, a hypothesis should be clearly formulated.
Response 6: Thank you for your suggestion. We agree that clearly formulating a hypothesis in the introduction section will enhance the clarity and focus of our study. We have revised the introduction to include a clear hypothesis:
It is hypothesized that LPT consumption, regardless of aging duration, will demonstrate significant improvements in metabolic outcomes, including blood pressure, lipid profiles, and glucose regulation, while simultaneously reducing body fat accumulation and reshaping gut microbiota composition.
We have supplemented the hypothesis in the background section. (Page2 Line85-89)
Comments 7: Since authors has conducted multi-group comparison, the level of statistical analysis should be mandatory adjusted, such as using Bonferroni or other similar tests.
Response 7: Thank you for your kind comments. For comparisons between different aging durations (1 year, 4 years, 7 years, and 10 years) of Liupao tea groups, we used one-way analysis of variance (ANOVA) to assess whether there were significant differences in intervention effects among the groups. If the ANOVA results had shown significant differences (P < 0.05), we would have conducted post-hoc pairwise comparisons using Bonferroni correction or the Least Significant Difference (LSD) method to control the overall Type I error rate. However, in this study, the ANOVA results indicated no significant differences among the groups, so we did not perform further pairwise comparisons, as multiple comparison corrections were not necessary in this case. Regardless of the final results, to ensure the rigor and transparency of our statistical analysis, we have supplemented the methods section with this explanation. We appreciate your professional suggestions, which have helped us improve the clarity and completeness of our description. (Page6 Line243-245)
Comments 8: Since 6 participants have dropped, how authors managed this? Authors should consider an intent-to-treat analysis including the people who dropped, rather than considering only the completers.
Response 8: Thank you for your important suggestion regarding the handling of participants who dropped out. We understand the significance of ensuring the robustness and generalizability of our results through an intent-to-treat (ITT) analysis.
We have supplemented our analysis with an ITT approach, which includes all participants who were randomized into the study (Table S3). Missing data for participants who dropped out were handled using multiple imputation by chained equations (MICE), generating plausible values for the missing follow-up data based on available baseline information and other covariates. The ITT analysis showed that Liupao tea of different aging durations (1 year, 4 years, 7 years, or 10 years) significantly modulated blood pressure, lipid profiles, and blood glucose levels, and had a lowering effect on body weight and body fat. The primary outcomes remained consistent with those observed in the completers-only analysis, further supporting the robustness of our findings. We have included this ITT analysis in both the methods and results sections. We appreciate your suggestion, as this approach enhances the rigor of our statistical analysis. (Page6 Line245-247)
Comments 9: The discussion section may benefit from a different arrangement, thus to be divided in 4 subsections as follows:
- The main findings of the study and their comparison with previously published papers
- The strengths and limitations of the study
- The clinical implications of the findings
- The new directions for future research
Response 9: Thank you for your valuable suggestion regarding the reorganization of the discussion section. We have carefully considered your advice and reorganized the discussion into three subsections, as follows:
4.1 Main findings and comparison with prior studies
4.2 The strengths and limitations of the study
4.3 Clinical implications and future directions
This revised structure effectively addresses your suggestion, while also ensuring a logical flow of ideas and a clearer presentation of the findings and their implications. We believe that this adjustment enhances the clarity and impact of the discussion, aligning it with the key elements you recommended. (Page12-15 Line383-524)
Comments 10: In the Authors’ contributions section please use the abbreviations of the authors.
Response 10: Thank you for your suggestion. We have revised the Authors' Contributions section to use abbreviations for the authors, ensuring consistency and clarity. We appreciate your feedback and have made the necessary adjustments to enhance the manuscript's professionalism. (Page15 Line540-544)
Round 2
Reviewer 1 Report
Comments and Suggestions for Authors
The authors have improved the manuscript, and solved every doubt.
Reviewer 2 Report
Comments and Suggestions for Authors
I accept the article after incorporating the revisions based on the reviewer's comments. The implemented changes have significantly improved the clarity and precision of the presented issues, as well as refined key substantive aspects. The reviewer's suggestions have been thoroughly addressed, enhancing the scientific value of the work. In its current form, the article meets the required standards and is ready for publication.
Reviewer 3 Report
Comments and Suggestions for Authors